# Laser Synthesis of Cerium-Doped Garnet Nanoparticles

**DOI:** 10.3390/nano13152161

**Published:** 2023-07-25

**Authors:** Volodymyr Vasylkovskyi, Iryna Bespalova, Andrey Evlyukhin, Yuriy Zholudov, Iaroslav Gerasymov, Daniil Kurtsev, Denys Kofanov, Olena Slipchenko, Mykola Slipchenko, Boris Chichkov

**Affiliations:** 1Institute for Scintillation Materials of NAS of Ukraine, Nauky Ave. 60, 61072 Kharkiv, Ukraine; ganina@isma.kharkov.ua (I.B.); yarosgerasimov@gmail.com (I.G.); kurcev@isma.kharkov.ua (D.K.); phkofanov@gmail.com (D.K.); slipchenko@iqo.uni-hannover.de (M.S.); 2Institute of Quantum Optics, Leibniz University Hannover, Welfengarten 1, 30167 Hannover, Germany; evlyukhin@iqo.uni-hannover.de (A.E.); chichkov@iqo.uni-hannover.de (B.C.); 3Department of Biomedical Engineering, Kharkiv National University of Radio Electronics, Nauky Ave. 14, 61166 Kharkiv, Ukraine; yuriy.zholudov@nure.ua; 4Department of Metals and Semiconductors Physics, National Technical University “Kharkiv Polytechnic Institute”, Kyrpychova Str. 2, 61002 Kharkiv, Ukraine; olena.slipchenko@khpi.edu.ua

**Keywords:** laser ablation in liquid, laser melting in liquid, crystalline nanoparticles, garnet, scintillators

## Abstract

The application of a pulsed laser ablation technique for the generation of cerium-doped garnet nanoparticles in liquids is investigated. The morphological and optical properties of the obtained nanoparticles are demonstrated. Features introduced by the single crystals of Gd_3_Al_2.4_Ga_2.6_O_12_:Ce^3+^, Lu_3_Al_5_O_12_:Ce^3+^, and Y_3_Al_1.25_Ga_3.75_O_12_:Ce^3+^ from which the nanoparticles are generated, as well as the parameters of a liquid media on the garnet nanoparticle generation are experimentally studied using TEM and UV-Vis spectroscopy methods. It is shown how the size, shape, and internal structure of the nanoparticles are related to the external laser ablation conditions, as well as to the laser melting processes of NPs in the colloidal solutions. This work provides important information about the generated nanoparticles, which can be used as building blocks for specially designed structures with predetermined optical properties.

## 1. Introduction

Garnets are a class of inorganic materials and have been known for many decades. The general chemical formula of garnet is A_3_B_2_C_3_O_12_, where A is a rare earth cation that occupies a site in the dodecahedron, B is a cation that occupies a site in the octahedron, and C is a cation that is located on a tetrahedral site. B and C atoms are coordinated by eight and six oxygen atoms, respectively [1,2]. Their cubic structure can offer the possibility of doping, a controllable host composition, good optical transparency, and chemical stability. The adjustable properties of garnet materials are due to a flexible cation substitution in the crystal lattice (Figure 1). Rare-earth-doped multicomponent garnet materials have been widely investigated and used in technological applications for decades. The properties of garnet structure also enable the preparation of transparent ceramics with low self-absorption and optical scattering [3]. Garnet materials are usually applied for fast scintillation detectors in the form of single crystals or ceramics for utilization in industrial and scientific applications. Cerium-doped garnet single crystals can be used for the following purposes: nuclear medical imaging, security purposes, astronomy, laser materials, and other research and development purposes [4,5,6].

In the past decades, there has been a fast expansion of nanotechnology and nanomaterials research, as well as their application in science and industry, which have raised the global demand for new advanced nanomaterials. Garnets in the nanoscale state usually feature higher luminescence intensity because of the light scattering and can offer controllable emission wavelength and improved efficiency depending on the size and type of the garnet material [7]. Ce-doped garnets in the form of nanoparticles (NPs) have prospects for applications in fast scintillation detectors with rapid emission response and also in light-emitting diodes (LEDs). The usage of nanosized garnet crystal particles can offer manufacturing flexibility and provide prospects for the development of novel LEDs by encapsulating them in an epoxy or silicone resin using a 3D printing technique and their following placement above electronic chips. Blue-emitting LEDs with Ce-doped luminescent garnet phosphor nanopowders can be exploited to make white light, because of good blue light absorption, which is subsequently converted into yellow light [8]. The development of garnet-based LED devices can allow for better control of light scattering and the minimization of optical losses [9]. Garnet NPs can also be used for composite scintillation materials production in combination with such technologies as spin coating, thermal polymerization, chemical bath deposition, inkjet printing, 3D printing, and ceramization [7,10,11].

Garnet NPs have been mostly fabricated using chemical methods like sol-gel, solvothermal, co-precipitation, hydrothermal, combustion, photo-induced synthesis, etc. [8,12]. Chemical techniques involve precursors to the control size and shape by guiding the growth kinetics of the NPs [13,14], but the application of chemical techniques for the NPs formation has several disadvantages. Chemical methods require a long processing time and the organization of large-scale production is difficult. The use of organic surface-blocking reactants, and other precursors for NPs formation, can be toxic and negatively influence the properties of colloidal solutions, which can lead to aggregation, catalyst deactivation, and background noise in analytical chemistry [3,14]. To avoid such negative effects, further extra-purification chemical steps may be required to eliminate surfactants and residual reactants.

To improve NP purity, scalability, and reproducibility of the synthesis process, greater attention has been devoted to the research and development of physical synthesis methods. The application of the laser ablation technique to generate garnet NPs can avoid the disadvantages of chemical synthesis techniques and conforms to the principles of “green chemistry”. In addition, because molecular precursors are not required, the laser ablation technique is more cost effective compared to the chemical ones. Laser synthesis of nanomaterials can be classified into the following processes: ablation, fragmentation, and melting. Laser ablation is commonly used for nanomanufacturing, microstructuring, and cutting of materials and is often described as a high-temperature and high-pressure process. The laser generation of NPs has already become a separate scientific direction. Laser ablation can be performed in a vacuum, gases and liquids. The liquid is a relatively safe medium for the laser synthesis of NPs, compared to synthesis in an air environment where workplace contamination may endanger personnel. Pulsed laser ablation in liquids (PLAL) is a fast and efficient physical top-down technique where the use of liquids as a medium for the laser synthesis of nanomaterials makes it possible to control the efficiency of ablation by lowering the heat load on the target material and increasing the shock pressure on the surface. NPs synthesized using the PLAL technique are often preserved with the same chemical composition as the target bulk materials. Generally, a part of the ablated NPs is oxidized or reduced during PLAL, forming surface defects that can provide electronic stabilization. Such volume or surface defects can also be beneficial for optical applications [15].

The PLAL technique requires laser setup, a target material in the form of bulk material or powder, and a liquid media. The size of the obtained NPs can be controlled using laser irradiation settings and parameters of the liquid media. Nowadays, most lasers have tunable parameters (pulse duration, repetition rate, and laser fluence) that often affect each other. The wavelength of the laser beam for PLAL can vary between ultraviolet to near-infrared, depending on the transparency of the liquid medium. Pulse duration may be in the range from femtoseconds to milliseconds. Laser pulse repetition rate, by shortening and lengthening the time between pulses, affects the number of pulses per unit of time, the average delivered energy, and the pulse fluence. When the pulse energy is above the fragmentation threshold, smaller particles are produced due to explosion and evaporation, whereas when the pulse energy is below the fragmentation threshold, the NP’s size can increase due to melting processes [15]. This effect was confirmed by nanosecond lasers, in which the heating-melting mechanism is responsible for the size variation of the NPs [16]. During PLAL, after the laser beam is absorbed by the target material and depending on the material and liquid media properties, the NPs are ejected using a cavitation bubble in the form of particles and crystalline debris (Figure 2).

It should be mentioned that during the PLAL generation of NPs, the laser beam can interact with the already generated NPs. The laser beam interacts with the NPs not only at the focus point, where laser ablation or pulsed laser fragmentation in liquids (PLFL) usually occurs, but also in other areas of the beam where the pulse power is several times lower. An additional laser melting of NPs can occur in those regions of the beam with low intensity. Such an effect is called pulsed laser melting in liquids (PLML) which has several differences from PLAL and PLFL. For PLAL and PLFL, ultrashort laser pulses with high energy densities (over several J × pulse^−1^ × cm^−2^) are used for the creation of explosive interactions with target bulk materials or particles for “top-down” particle production. On the other hand, for the PLML synthesis of nano- (1–100 nm) and submicrometer (100 nm–1 μm) particles, laser pulses with energy densities one or two orders of magnitude lower than for PLAL and PLFL are used (10–100 mJ × pulse^−1^ × cm^−2^). PLML is sometimes considered a modified PLAL technique with low-energy laser irradiation. In PLML, laser pulse interaction with particles leads to the aggregation of NPs and subsequent formation of larger particles, calling this method a “bottom-up” technique. As a result of an appropriately executed PMLM process, sphere-shaped particles are produced. But due to the uneven distribution of power within the beam, partially melted nanoparticles and agglomerates that consist of fused nanoparticles can also be generated (Figure 3). The PLML technique can be applied for the spheroidization of particles in colloids or substrate modification. For the PLML generation of NPs, raw materials, particles obtained using PLAL, PLFL, and other physical and chemical techniques can be used [17].

Until now, laser techniques have been applied to fabricate a variety of nanomaterials including noble metals, semiconductors, alloys, and, to a smaller degree, for the generation of multicomponent crystalline NPs (binary, ternary, and quad materials) [17,18]. The application of PLAL and PLML can make it possible to obtain garnet NPs with new properties and pure NP surfaces. A better understanding of the laser fabrication mechanisms of multi-component garnet nanomaterials is needed to control the size and shape of the NPs. Also, in order to understand the preservation or loss of material properties, it is necessary to pay attention to the crystallinity of the generated nanostructures.

## 2. Materials and Methods

### 2.1. Laser Setup

The laser generation of NPs was conducted using the experimental laser setup, illustrated in Figure 4. A nanosecond diode-pumped laser Alphalas PULSELAS P–355–100–HP and a fiber-coupled laser diode LDF–45–P were used. Single crystals as target materials were placed in a quartz cuvette and fixed with a homemade metal holder. The cuvette was filled with a liquid medium (chloroform or sodium citrate water solution). Since the optimal liquid thickness for the maximum synthesis productivity, between the cuvette wall and the target materials’ surface, is 1–5 mm, the surface of the target material was positioned at a distance of 5 mm from the transparent cuvette wall used for laser beam incidence. The laser beam was pointed through the transparent cuvette wall at the surface of the target materials using a mirror with a coated side and a converging lens with a 60 mm focus. Since the laser system was operated without a laser scanner, a handmade cuvette manipulator was used to move the cuvette with the target material, with respect to the laser beam to conduct uniform layer-by-layer ablation of single crystals. The translational circular motion of the cuvette manipulator was regulated by the power supply (the voltage of the power supply varied from 9 to 11 V, which affected the rotation rate) and adjusted to a circle with a 5 mm diameter.

### 2.2. Liquid Media

Liquid parameters are crucial for the PLAL formation of NPs because of their influence on the size and mobility of cavitation bubbles, as well as on the local concentration of colloidal solution [15]. PLAL in aqueous solutions of monovalent or divalent ions can lead to the formation of elongated NPs or NP chains [19]. Also, the viscosity of the liquid media can affect the NP’s stability and synthesis productivity, which can be explained by the formation of persistent bubbles, the dwell time of which depends on the liquid’s viscosity. However, it should be noted that after the interaction of the laser beam with liquid media, decomposition products of the solvent may appear in the colloidal solution. For example, our previous studies have shown that after PLAL in toluene, dark micro-sized flakes appear, which are presumably composed of carbon. In this work, chloroform (99.8%, Carl Roth GmbH + Co. KG, Karlsruhe, Germany) without additional purification and a sodium citrate water solution were used as the liquid media for PLAL (Table 1). Sodium citrate water solution, with the concentration of C = 1.6 × 10^–3^ M, was obtained by dissolving a sample of Na_3_C_6_H_5_O_7_ × 3H_2_O in distilled deionized water (Laboratory ultra-pure water purification unit – Milli-Q Integral system, Merck Millipore, Darmstadt, Germany). Sodium citrate water solution was chosen and used as a capping agent to prevent NP aggregation. Chloroform was chosen as a liquid medium for PLAL, taking into account the future potential applications in LEDs and scintillation materials in combination with 3D printing and other polymerization techniques. It is easier to incorporate garnet NPs, generated and stored in chloroform, into polymers that dissolve in chloroform like polymethyl methacrylate or polyvinyl butyral.

### 2.3. Garnet Single Crystals

For garnet NP generation, as target materials, single crystals of gadolinium aluminum gallium garnet doped with cerium (GAGG:Ce), lutetium aluminum garnet doped with cerium (LuAG:Ce), and yttrium aluminum gallium garnet doped with cerium (YAGG:Ce) were used. Garnet single crystals were grown from melts with the following stoichiometric compositions: Gd_3_Al_2.4_Ga_2.6_O_12_: Ce^3+^ (0.3 at. % Ce), Lu_3_Al_5_O_12_:Ce^3+^ (0.5 at. % Ce), and Y_3_Al_1.25_Ga_3.75_O_12_:Ce^3+^ (0.3 at. % Ce). Starting oxides Y_2_O_3_, Gd_2_O_3_, Lu_2_O_3_, Al_2_O_3_, and Ga_2_O_3_ had a purity no worse than 4N (99.99%). Powders for crystal growth were synthesized using a solid-state reaction under a temperature above 1600 °C in a neutral atmosphere for 10 h. Single crystals were fabricated using the Czochralski technique in an induction heating furnace “OXIDE” (Institute for Scintillation Materials of NAS of Ukraine, Kharkiv, Ukraine). GAGG:Ce and YAGG:Ce garnets that contained gallium were grown from iridium crucibles with a diameter of 60 mm and height of 60 mm under a weakly oxidizing atmosphere. LuAG:Ce was grown from a tungsten crucible with a diameter of 50 mm and height of 50 mm under an Ar + CO atmosphere. Growth parameters, such as pulling rate and rotation rate for each of the growth processes, were in the ranges of 1–1.5 mm/h and 10–15 rpm, respectively. As seed crystals, crystal fragments with crystallographic orientation relevant to (100) with square sections of 5 × 5 mm^2^ were used. To eliminate thermal stress after growth, single crystals were cooled down to room temperature during the 15 h wait and then annealed at temperatures above 1400 °C in an air atmosphere for 10 h. One side of every single crystal was polished for the homogeneous motion of the focused laser beam over the surface of the target material. Samples of each grown crystal are presented in Figure 5. More details about crystal growth are described in [22,23,24].

### 2.4. Characterization of Garnet NPs

Laser-generated NPs were characterized using transmission electron microscopy (TEM) and UV-Vis absorption spectroscopy. High-resolution images and electron-diffraction patterns were measured with an analytical TEM “Tecnai G^2^ F20 TMP” from FEI (Laboratory of Nano and Quantum Engineering, Leibniz University Hannover, Hannover, Germany). The UV-Vis absorption spectra were measured in quartz cuvettes using the “Shimadzu UV–1900i” double beam spectrophotometer (Institute of Quantum Optics, Leibniz University Hannover, Hannover, Germany).

## 3. Results and Discussion

For the laser generation of the NPs from every garnet single crystal, the laser setup was operated using the following parameters: λ ≈ 1064 nm; Δt ≈ 845 ns; repetition rate ≈ 560 Hz, pulse energy ≈ 212 µJ, with vertical polarization of the laser beam. For obtaining sufficiently concentrated colloidal solutions (Figure 6, bottom right insert), the most suitable synthesis time was 30 min. With a longer duration of PLAL, in some cases, the obtained over-saturated colloidal solution can absorb irradiation and block laser beams from reaching the single crystals’ surface.

The TEM measurements of the colloidal solutions were conducted without a concentration decrease (Figure 6). The estimated size distributions of the NPs (Figure 6a–c) and the average size of the NPs generated in chloroform were calculated from the acquired TEM images: 19.8 nm for GAGG:Ce, 24 nm for LuAG:Ce, and 22.4 nm for YAGG:Ce. The observed difference in the average size of nanoparticles can be directly related to the melting point of the target single crystals: the higher the melting point of the target material (Table 2), the bigger the average size of the PLAL-synthesized NPs. The dependence of the average size of the laser-generated garnet NPs on the melting point of single crystals is illustrated below (Figure 7).

According to the TEM observations, the colloidal solutions mostly contained partially spherically-shaped NPs (Figure 6) and, to a lesser extent, spherically-shaped NPs (Figure 8) and crystal debris (Figure 9). It is assumed that the presence of partially spherically-shaped NPs and spherically-shaped NPs can be attributed to the formation of NPs from partially-melted and completely melted crystal fragments that were ejected from the target material. The generation of melted and partially-melted NPs can be attributed to the “secondary” PLML effect where laser irradiation interacted with material ejected from single crystals in the form of NPs using the PLAL process. Partially-melted NPs were obtained from every garnet single crystal in both the liquid media and spherically-shaped NPs were obtained only from GAGG:Ce in chloroform (Figure 8). Also, the partially-melted NPs were agglomerated into clusters with an approx. size of 50 nm (Figure 6a,b,e). The appearance of agglomerates consisting of several nanoparticles (≈4–10) may also be the result of an incomplete PLML process, in which the temperature was not sufficient for the formation of spherical particles. It should be noted that signs of the partially-melted NPs crystallinity were observed (Figure 6d), which means that the obtained NPs have partially retained their crystalline structure. At the same time, in the spherical particles, no signs of the presence of a crystalline structure were noticed, presumably due to the formation of an amorphous structure from the completely molten material.

Colloidal solutions also contained debris with various shapes and approximate sizes of more than 50 nm (Figure 9), which were presumably knocked out by a micro-explosion on the surface of the single crystal during PLAL. The presence of signs of a crystal lattice in the obtained crystal debris was observed (Figure 9, top left insert) but the overall degree of crystallinity of the obtained NPs, based on the available data, is unknown and will be a subject of further research. The TEM images of the crystal debris show no signs of melting processes of the material. Presumably, due to their large size, the crystal debris were not stable in the colloidal solution and quickly sedimented to the bottom of the cuvette, so they could not be repeatedly exposed to the laser irradiation.

The UV-Vis absorption spectra of the colloidal solutions obtained and stored in chloroform and the sodium citrate water solution are shown in Figure 10. The UV-Vis absorption of the colloidal solutions in chloroform is several times higher than in the sodium citrate water solution and can be related to the higher concentration of NPs in the chloroform colloidal solution (that was also noticed during the TEM measurements). Compared to the absorption spectra of garnet single crystals, the 350 nm and 450 nm peaks related to Ce dopant are absent for the laser-generated colloidal solutions. The 274 nm peak for GAGG:Ce NPs obtained in the sodium citrate water solution can be related to the presence of Gd [27,28] and are not noticeable in chloroform.

## 4. Conclusions

A comprehensive study of laser generation of garnet NPs has been conducted. Garnet NPs have been generated using the laser technique from Gd_3_Al_2.4_Ga_2.6_O_12_:Ce^3+^ (GAGG:Ce), Lu_3_Al_5_O_12_:Ce^3+^ (LuAG:Ce), and Y_3_Al_1.25_Ga_3.75_O_12_:Ce^3+^ (YAGG:Ce) single crystals that were grown using the Czochralski technique. Pulsed-laser-ablation generation was conducted in chloroform and a sodium citrate water solution. The synthesized garnet NPs have been characterized using transmission electron microscopy and UV-Vis absorption spectroscopy. All characterization methods indicated a less efficient laser generation in the sodium citrate water solution than in chloroform. Characterization of the morphology of the obtained garnet NPs showed signs of additional interaction of laser radiation with nanoparticles in a colloidal solution, which led to the formation of spherical particles, partially molten particles, and agglomerates of nanoparticles via the pulsed laser melting process. It has been observed that liquid media parameters have an impact on the NP’s shape and concentration in the colloidal solutions. A dependence of the average size of the synthesized NPs on the melting point of garnet single crystals has been observed. TEM images and diffraction patterns have shown that crystallinity is preserved in crystal debris, partially molten NPs, and agglomerates, but is absent in spherically-shaped NPs formed from completely molten crystal fragments. The UV-Vis absorption spectra of the NP colloidal solutions have shown an absence of Ce in all garnet NPs and the presence of Gd in GAGG:Ce NPs compared to the bulk materials.

## Figures and Tables

**Figure 1 nanomaterials-13-02161-f001:**
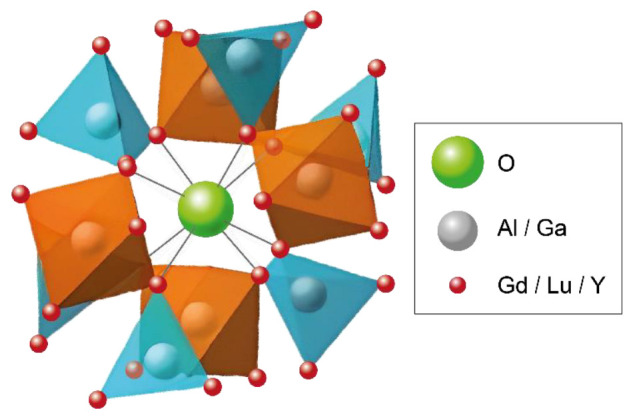
Crystal structure of garnet.

**Figure 2 nanomaterials-13-02161-f002:**
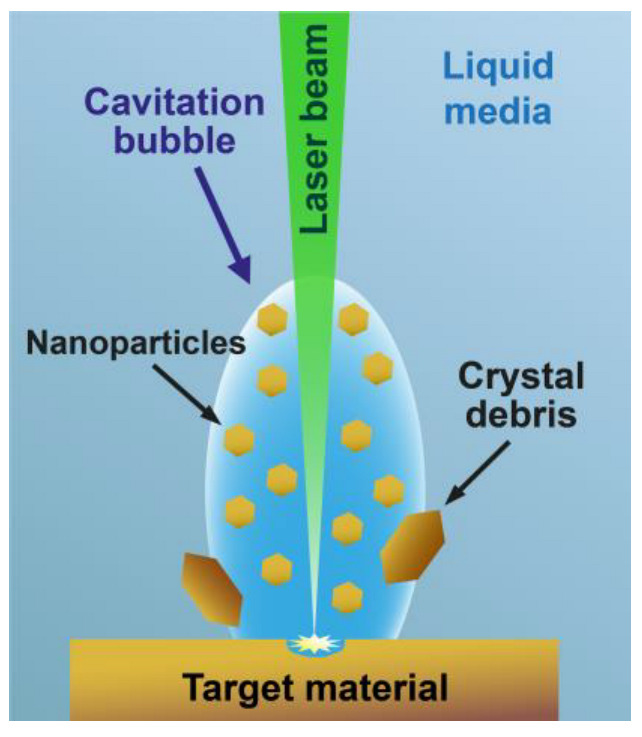
Scheme of laser generation of NPs by pulsed laser ablation in liquids technique.

**Figure 3 nanomaterials-13-02161-f003:**
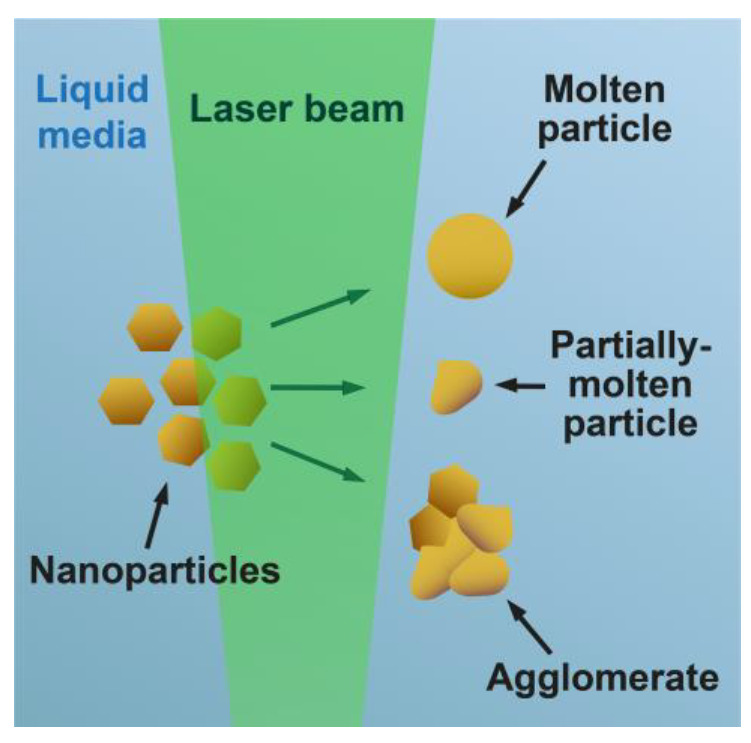
Scheme of laser transformation of NPs using pulsed laser melting in liquids technique.

**Figure 4 nanomaterials-13-02161-f004:**
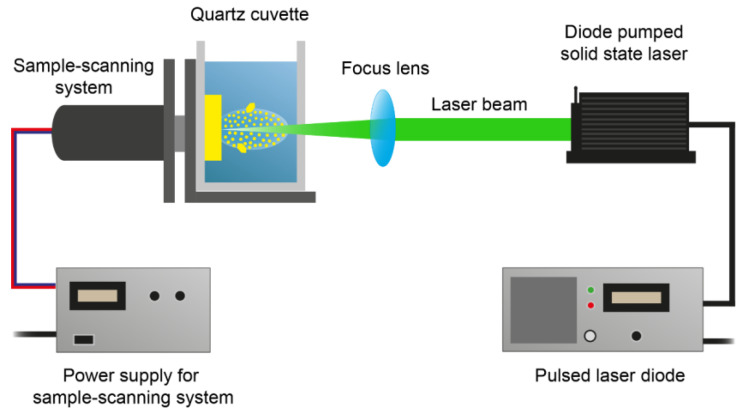
Scheme of setup for the laser generation of NPs.

**Figure 5 nanomaterials-13-02161-f005:**
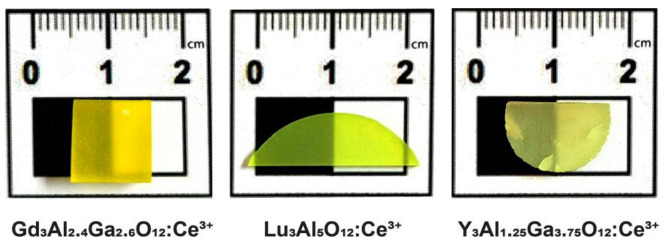
Garnet single crystals used as target materials for laser generation of NPs: Gd_3_Al_2.4_Ga_2.6_O_12_:Ce^3+^ (GAGG:Ce), Lu_3_Al_5_O_12_:Ce^3+^ (LuAG:Ce), and Y_3_Al_1.25_Ga_3.75_O_12_:Ce^3+^ (YAGG:Ce).

**Figure 6 nanomaterials-13-02161-f006:**
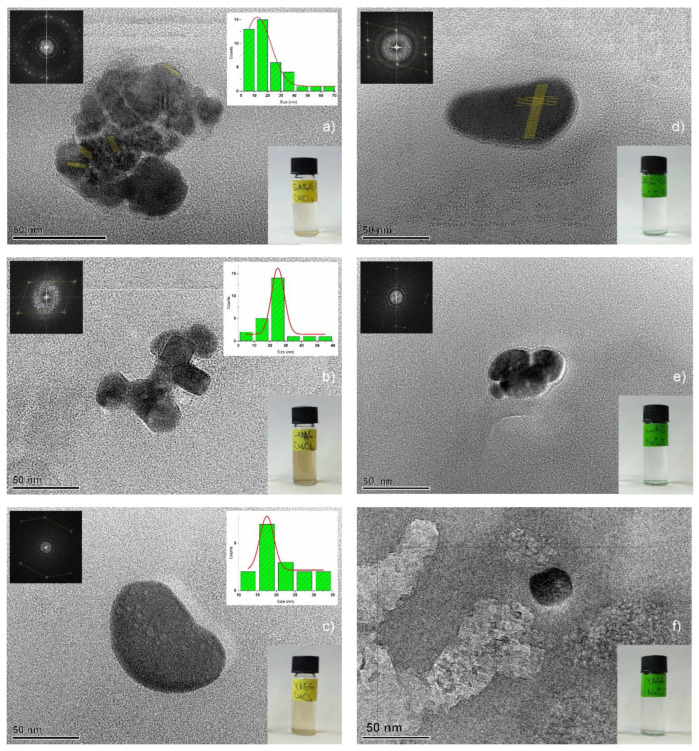
TEM images of garnet NPs obtained using PLAL (top left insert shows the pattern of electron diffraction, top right insert shows the NPs’ size distribution, bottom right insert shows corresponding photos of colloidal solutions) generated in chloroform: (**a**) GAGG:Ce; (**b**) LuAG:Ce; (**c**) YAGG:Ce, generated in sodium citrate water solution: (**d**) GAGG:Ce; (**e**) LuAG:Ce; and (**f**) YAGG:Ce.

**Figure 7 nanomaterials-13-02161-f007:**
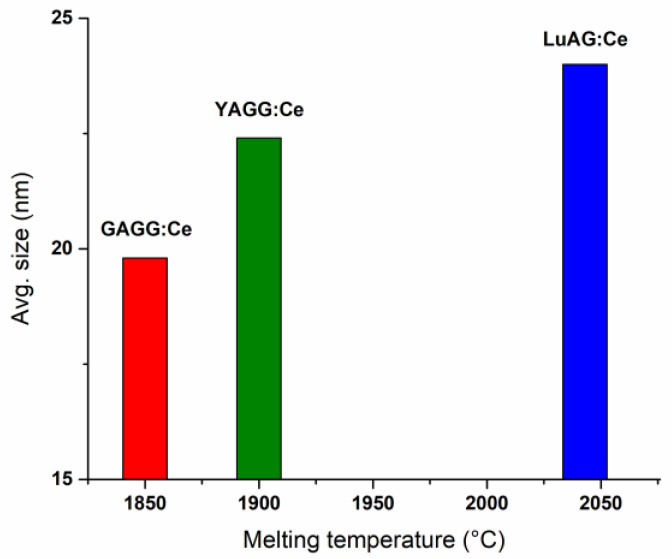
Dependence of the average size of garnet NPs, generated by PLAL in chloroform, on the melting temperature of single crystals.

**Figure 8 nanomaterials-13-02161-f008:**
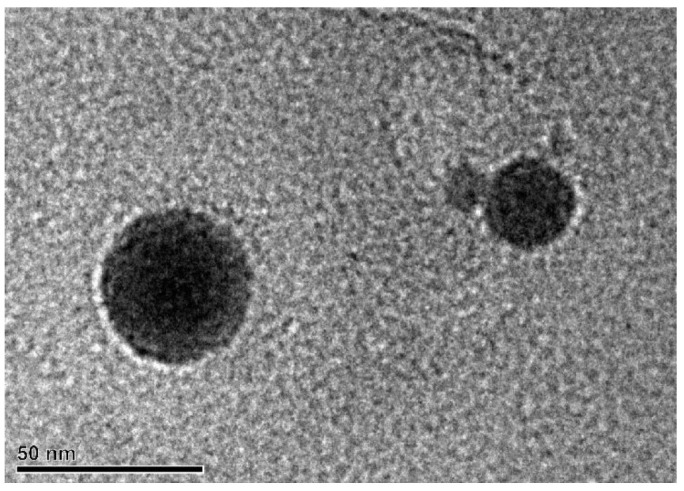
TEM image of spherically-shaped NPs presumably obtained by PLML from GAGG:Ce in chloroform.

**Figure 9 nanomaterials-13-02161-f009:**
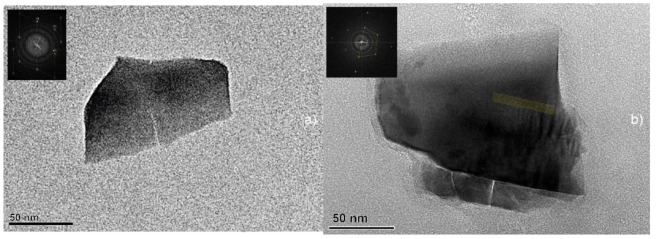
TEM images of crystal debris of garnet crystals obtained as a result of PLAL (top left insert shows the electron diffraction): (**a**) GAGG:Ce generated in sodium citrate water solution and (**b**) LuAG:Ce generated in chloroform.

**Figure 10 nanomaterials-13-02161-f010:**
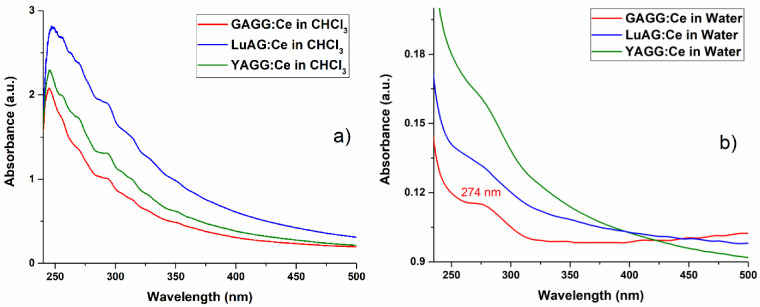
UV-Vis absorption spectra of colloidal solutions with garnet NPs synthesized and stored in chloroform (**a**) and in sodium citrate water solution (**b**).

**Table 1 nanomaterials-13-02161-t001:** Parameters of liquid media for PLAL synthesis of garnet NPs.

Medium	Thermal Conductivity (20 °C)	Boiling Point	Density	Viscosity (20 °C)	Refractive Index (280 nm, 20 °C)
Chloroform	0.129 W/m·K	61 °C	1.48 g/cm^3^	0.563 mPa·s	1.498 [20]
Sodium citrate water solution	≈0.598 W/m·K	≈100 °C	≈1 g/cm^3^	≈1.0016 mPa⋅s	1.361 [21]

**Table 2 nanomaterials-13-02161-t002:** Parameters of garnet single crystals.

Single Crystal	Melting Temperature	Density
GAGG:Ce	1850 °C [25]	6.63 g/cm^3^ [22]
LuAG:Ce	2043 °C [26]	6.23 g/cm^3^ [23]
YAGG:Ce	Approx. 1900 °C	up to 5.8 g/cm^3^ (depending on Ga^3+^ concentration) [24]

## Data Availability

Data available on request.

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
