# Peer review of "Laser Synthesis of Cerium-Doped Garnet Nanoparticles"

_nanomaterials, 2023, doi:10.3390/nano13152161_

Round 1

Reviewer 1 Report

The article “Laser synthesis of cerium-doped garnet nanoparticles” Vasylkovskyi et.al devoted to formation of nanoparticles by laser ablation in liquids. The novelty is generation of new type nanoparticles. Please see some comments below.

1. Typo: The usage of precursors for NPs formation can negatively influence on properties of NPs and further purification steps may be required.

2. How you explain the choice of liquids?

3. How the chemical compound of liquid influence on NPs morphology? For example laser ablation of targets in monovalent or divalent aqueous solutions leads to formation of elongated nanoparticles or NPs chains [J.-P. Sylvestre, S. Poulin, A.V. Kabashin, E. Sacher, M. Meunier, J.H.T. Luong, J.Phys. Chem. B 108 (2004) 16864, Zhilnikova, M. I., Barmina, E. V., Shafeev, G. A., Pridvorova, S. M., & Uvarov, O. V. (2020). Laser-assisted generation of elongated Au nanoparticles in aqueous solutions of divalent ions. Gold Bulletin, 53, 129-134.]

4. For PLAL generation of garnet NPs, a sub-nanosecond diode-pumped laser Alphalas 67 P-355-100-HP was used. But after you say that laser beam had pulse width of 845 ns. You used sub-nanosecond laser or nanosecond? Also indicate pulse energy and intensity of the laser beam on the target surface.

5. Captions of Fig. 3 and Fig. 4 are the same. Please check.

6. It is possible to present TEM general view of nanoparticles? It is not clear how you did NPs size distribution.

7. It would be nice to present absorption spectra from 250 to 800-900 nm.

8. If it is possible to see XRD results?

9. Typo: A Dependence of the average size of synthesized NPs on the melting point of single crystals has been observed.

10. Reference #21 cited by Ukrainian language.

Author Response

  1. Typo: The usage of precursors for NPs formation can negatively influence on properties of NPs and further purification steps may be required.

The changes are added.

  1. How you explain the choice of liquids?

Sodium citrate water solution was chosen and used as a capping agent to prevent NPs aggregation. Chloroform was chosen as a liquid medium for PLAL taking into account the future potential applications in LEDs and scintillation materials in combination with 3D printing and other polymerization techniques. It is easier to incorporate garnet NPs, generated and stored in chloroform, into polymers that dissolve in chloroform like polymethyl methacrylate or polyvinyl butyral.

Corresponding additions have been made to the article.

  1. How the chemical compound of liquid influence on NPs morphology? For example laser ablation of targets in monovalent or divalent aqueous solutions leads to formation of elongated nanoparticles or NPs chains [J.-P. Sylvestre, S. Poulin, A.V. Kabashin, E. Sacher, M. Meunier, J.H.T. Luong, J.Phys. Chem. B 108 (2004) 16864, Zhilnikova, M. I., Barmina, E. V., Shafeev, G. A., Pridvorova, S. M., & Uvarov, O. V. (2020). Laser-assisted generation of elongated Au nanoparticles in aqueous solutions of divalent ions. Gold Bulletin, 53, 129-134.]

Thank you for the information, additions have been made to the article.

  1. For PLAL generation of garnet NPs, a sub-nanosecond diode-pumped laser Alphalas 67 P-355-100-HP was used. But after you say that laser beam had pulse width of 845 ns. You used sub-nanosecond laser or nanosecond? Also indicate pulse energy and intensity of the laser beam on the target surface.

The changes are added: the nanosecond laser was used with pulse energy ≈ 212 µJ.

  1. Captions of Fig. 3 and Fig. 4 are the same. Please check.

Thank you for your comment. Captions have been corrected.

  1. It is possible to present TEM general view of nanoparticles? It is not clear how you did NPs size distribution.

The size distribution of the obtained nanoparticles was estimated from several TEM images. An example for GAGG:Ce generated in chloroform is illustrated below.

  1. It would be nice to present absorption spectra from 250 to 800-900 nm.

Since the signal of the absorption spectrum becomes flat after 500 nm and does not carry useful information, and due to the fact that the necessary information about the material (absorption peaks associated with Gd and Ce) is in the region of 200 - 500, this measurement range was chosen.

  1. If it is possible to see XRD results?

Unfortunately, the power of our laser does not allow us to obtain a sufficient amount of material for XRD studies, therefore our article is focused on studying the morphological properties of NPs after the process of laser ablation and melting of nanocrystals.

  1. Typo: A Dependence of the average size of synthesized NPs on the melting point of single crystals has been observed.

The changes are added.

  1. Reference #21 cited by Ukrainian language.

The reference is translated

Reviewer 2 Report

In this paper, laser ablation techniques were used to ablate material suspensions to obtain nanoscale particles with different characteristics and sizes. However this paper suffers from the following problems:

1.What is the mechanism of action of the laser on the material? What is the reason for the generation of nanoscale particles?

2.TEM is not able to fully characterise the material and a variety of test methods are required to investigate the various properties of crystalline materials.

The grammatical presentation of this article is clear and precise.

Author Response

  1. What is the mechanism of action of the laser on the material? What is the reason for the generation of nanoscale particles?
  • During PLAL, after the laser beam is absorbed by the target material, depending on the material and liquid media properties, NPs are ejected by a cavitation bubble in the form of particles, and crystalline debris. After NPs ejection into liquid, laser beam can interact with already generated NPs that leads to laser melting of nanomaterials and subsequently to generation of spherical particles and agglomerates.
  • Garnets in the nanoscale state usually feature higher luminescence intensity and can offer controllable emission wavelength and improved efficiency depending on the size and type of the garnet material. Ce–doped garnets in the form of nanoparticles (NPs) have prospects for applications in fast scintillation detectors with rapid emission response and also in light–emitting diodes (LEDs). The usage of nanosized garnet crystal particles can offer manufacturing flexibility and provide prospects for the development of novel LEDs by encapsulation in an epoxy or silicone resin via 3D–printing technique, and their following placement above electronic chips. Garnet NPs can also be used for composite scintillation materials production in combination with such technologies as spin–coating, thermal polymerization, chemical bath deposition, inkjet printing, 3D printing, and ceramization. A part of laser-processed NPs is oxidized or reduced during PLAL, forming surface defects that can provide electronic stabilization. Such volume or surface defects can also be beneficial for optical applications.

Corresponding additions have been made to the article.

  1. TEM is not able to fully characterize the material and a variety of test methods are required to investigate the various properties of crystalline materials.

Our article is more focused not on the composition of the resulting material, but on morphological properties after the processes of particle formation. For our work, it was important to know the information about the presence or absence of a crystal structure, which was possible to do using the TEM method.

Round 2

Reviewer 1 Report

Good morning, 

thank you very much for your answers to my questions.